# Spatiotemporal Economic Analysis of Corn and Wheat Production in the Texas High Plains

**Aminun Naher** [1], **Lal K. Almas** [1,*], **Bridget Guerrero** [1] and **Sania Shaheen** [1,2]

[1] Paul Engler College of Agriculture and Natural Sciences, West Texas A&M University, Canyon, TX 79016, USA; bonoecon@gmail.com (A.N.); bguerrero@wtamu.edu (B.G.); saniaashaheen@gmail.com (S.S.)
[2] International Institute of Islamic Economics (IIIE), Islamic International University, Islamabad 44000, Pakistan
[*] Correspondence: lalmas@wtamu.edu

**Abstract:** The aim of this study is to visualize the historical changes in wheat and corn cropping patterns in the Texas High Plains from the perspective of geographical concentration and spatial autocorrelation. Historical county-level agricultural census data were collected from the United States Department of Agriculture and the National Agricultural Statistics Service from 1978 to 2017. Exploratory data analysis techniques were employed to examine the geographical concentration and spatial dependence of crop production among nearby locations. The results of temporal changes indicate that the harvested acres of corn and wheat tended to decrease throughout the study period. Total and irrigated harvested corn and wheat acreages were concentrated in a smaller number of counties over time while wheat production was mainly concentrated in the northern part of the region. The Moran's I test statistic for total and irrigated areas of cropland suggest that there was spatial dependence among the neighboring counties in crop production in this region. In summary, there was a spatiotemporal change in cropping patterns in the Texas High Plains over the study period. Based on the results of the spatiotemporal changes in cropping patterns in the Texas High Plains, policy makers should promote and support non-irrigated varieties of crops in order to decrease the dependence on irrigation water from the Ogallala Aquifer.

**Keywords:** cropping pattern; Moran's I test statistic; Ogallala Aquifer; spatiotemporal change; Texas High Plains





## 1. Introduction

The Texas High Plains is one of the most extensive agricultural areas in the United States. This region covers about 37,676 square miles and is comprises 39 counties. Agriculture is one of the major economic drivers in this region, which is a hub for substantial agriculture production in the state of Texas. In 2017, there were 29,360,229 acres of cropland in Texas of which 6,595,607 acres (78%) were contained in the 39 counties of the Texas High Plains [1].

The history of agriculture involves the human-induced spatial movement of crop production. The agricultural industry in this region faces unique agronomic, environmental, and economic challenges due to extreme weather conditions and water scarcity. The climate is semi-arid, and climatic changes are leading to reduced regional rainfall and increased crop water demand. Therefore, irrigated agriculture production in this region faces great threat [2]. Due to low rainfall rates, most of the agricultural producers in the region rely on the Ogallala Aquifer for irrigation water [3]. The Ogallala Aquifer was formed by ancient runoff from the Rocky Mountains. This aquifer was first discovered by the United States Geological Survey (USGS) in the 1890s. After World War II, farmers used large-scale irrigation technologies to extract ground water from the Ogallala Aquifer [4].

The use of groundwater has been increasing for irrigation. However, according to the USGS, after 1974, the water tables have declined significantly from pre-development

levels [5]. In 1978, the total harvested area of irrigated cropland was 4,393,257 acres in the Texas High Plains [6]. However, in 2017, this area had decreased to only 2,940,888 acres [1]. The main irrigated crops grown in this area are corn, cotton, sorghum, and wheat; cotton, sorghum, and wheat can be grown in both irrigated and non-irrigated areas. In 2017, the total irrigated acres harvested for corn, cotton, sorghum, and wheat were 519,029, 1,517,214, 98,708, and 236,879 acres, respectively. The availability of irrigated water and the temperate climatic conditions have made this area suitable for crop production. From 2012 to 2017, the Texas High Plains ranked third and fifth among other states in the U.S. for cotton and sorghum production, respectively. For the same period, the Texas High Plains also ranked 15th for corn and wheat production nationwide [7].

### 1.1. Spatial and Temporal Analysis in Agriculture

Understanding the spatial and temporal changes in the production of major crops in a specific area is important for effective, evidence-based agricultural and economic policies. Historical changes in geographical distribution and concentration of livestock production were examined in the United States. The results indicated that the greatest change in geographical concentration was in laying hen egg production and pullet inventory over the study period. The results also suggested that geographical concentrations in the other livestock industry sectors were not as high as those in broiler production [8]. Another study examined the historical change in the spatial movement of plants and animals from 1879 to 2007. The results of the study indicate that corn production in the United States increased dramatically during the twentieth century. The results also revealed that environmental, biological, and spatial changes play a vital role in crop production. There are also other factors such as soil type, elevation, rainfall, pests and disease, sunlight, and temperature that limit agricultural productivity [9].

Another study investigated the relationship between producers' crop price expectations and groundwater pumping decisions. County-level data were collected from Northwest Kansas Groundwater Management District 4 (GMD4) and monthly precipitation data were collected from the PRISM Climatic Group. Kansas monthly cash price data were used to construct expected crop prices from 1997 to 2016. This study focused on the five most common irrigated crops (namely alfalfa, corn, sorghum grain, soybean, and wheat). The estimated results suggested that producers of northwest Kansas adjust the quantity of groundwater pumped in response to changes in precipitation for various irrigated crops. However, there was no statistically significant relationship found between crop price expectations and groundwater pumping decisions [10].

### 1.2. Exploratory Spatial Data Analysis

The Geographical Information System (GIS) is a very powerful tool for visualizing spatial patterns. GIS-based maps can be created to explore historical patterns of crops, urbanization trends, land use or cover changes, and water use in industry. The change in production decisions or land use can be easily visualized from these maps. Several studies that have applied GIS-based approaches for exploratory spatial data analysis are reviewed in this section.

Reference [11] proposed a method that allows long-term mapping of cropping patterns using time-series crop maps. Crop maps were derived from supervised classification of remote sensing data. This study applied GIS overlay analysis operations to derive the spatial and temporal relationships between crops. The results of the study show that the application of the method to the study area revealed a large variability in cropping patterns. Guerrero et al. (2019) focused on the impact of dairy industry expansion on water use, crop composition, and the local economy. Data on dairy cow inventory and annual irrigated crop acres were collected from the Federal Milk Marketing Order and Farm Service Agency, respectively. The Wilcoxon test and SAS PROC NPERWAY methods were employed to determine the significant difference between the number of acres cultivated at the beginning of 2000 and the most recent data for 2015. Moran's I statistics were presented

to visualize the spatial autocorrelation between neighboring dairy industries in the Texas High Plains. The total irrigated area decreased by 17.8 percent from 2000 to 2015, indicating a trade-off between increased irrigation requirements due to dairy feed demand and the overall irrigation demand in the region. Moran's I statistics suggested that the spatial autocorrelation of dairy inventory by county in the study region experienced a positive increase from 2000 to 2015. The authors of [12] examined the urbanization trends of Hebei Province in China using the Geographical Information System (GIS) and remote sensing. The objective of the study was to explore the temporal and spatial characteristics of urban expansion and to examine land cover changes due to urbanization between 1987 and 2001. To achieve these objectives, multi-annual socio-economic statistics and two types of satellite multi-spatial images were collected from 1934 to 2001. GIS software (MapInfo5.0) was used to create maps of the urban area of Shijiazhuang City in different historical periods. The results indicated that the urban area of Shijiazhuang City expanded by 96% from 1934 to 2001. However, the annual growth rate varied significantly in different periods, and the fastest expansion stage was from 1981 to 2001. The results from the landscape change due to high-speed urbanization show that urban regions have increased sharply while agricultural land has decreased significantly.

*1.3. Cropping Patterns in the Texas High Plains*

Agriculture in the Texas High Plains is different from that in other areas of the United States. The Texas High Plains are a semi-arid region, and irrigation is vital to this region. The depletion of groundwater sources is a growing concern for crop production in this region. This section reviews studies related to cropping patterns in the Texas High Plains. The authors of [4,13] focused on the historical change in groundwater availability mainly from the Ogallala Aquifer, the short and long-term effects of agriculture's adaption to water resources, and the threat of drought. Data were collected from the census of agriculture and the United States geological survey, and the baseline model was used to assess the adaptation of groundwater in agricultural production. A placebo test was used to explore the local spillover effects throughout the nearby counties of the Ogallala Aquifer. Groundwater reduces the negative impact of drought on water-intensive crops. The results of the study indicated that from 1970 to 1997, irrigation increased by 11 percent for those counties that lie above the Ogallala Aquifer. Since corn is a water-intensive crop, the irrigated area of corn decreased throughout this time period in the study region. The authors of [14] examined the production levels and management practices of corn producers in the Texas High Plains with reduced or limited levels of irrigation. Corn has a high evapotranspiration (ET) demand (both daily and seasonally) in the Texas High Plains [15,16]. Although corn yield varied from year to year, there has been a clear linear upward trend from 1975 to 2015. Management practices are more important than breeding when water exists in limited conditions. Irrigation management is the most effective way to sustain high crop productivity. The authors stated that breeding for drought tolerance in corn is a major goal to improve yield stability under drought conditions. The results show that newly developed drought-tolerant corn hybrids provide yield benefits of 10–15 percent under limited (reduced) irrigation water levels. The results of the study suggest that management practices for irrigated corn in the Texas High Plains require proper management, hybrid selection, a high seeding rate, and planting date planning to achieve higher yields. Subsequently, the reference [17] analyzed that irrigated agricultural production in eight states relies on water from the Ogallala Aquifer (Southern South Dakota, Southeast Wyoming, Eastern Colorado, Nebraska, Western Kansas, Eastern New Mexico, Northwest Oklahoma, and Northwest Texas). Crop production data and weather information were collected from 1960 to 2007 for 205 counties from the abovementioned eight states. A regression technique was used to estimate the irrigation elasticity (IR), which is related to the ratio of county dry matter yields to county share of irrigated area. The results of the study showed that in 2007, most of the irrigated agricultural production was produced in Nebraska which was worth around USD two billion. Nebraska covers 36% of the total study area and accounts for 69%

of the total volume of water. Moreover, the results explain that Nebraska benefited most from water withdrawn for irrigation compared to Kansas and Texas.

Based on previous literature, very few studies have examined the spatiotemporal patterns of crop production in the Texas High Plains. Therefore, this research adds to the existing literature by analyzing county-level, time-series data on crop production. In particular, several exploratory data analysis techniques were employed to examine and visualize changes in spatiotemporal patterns of crop production in this region over the past 40 years.

Few studies have examined the spatiotemporal patterns of agricultural production in the Texas High Plains. In addition, most of these studies on the evolution of regional economic activity in this region have focused mainly on the impacts of one-time major events. To help regional producers and the public make better and more informed decisions, it is essential to have visuals that communicate complex information about the spatiotemporal dynamics of changes in regional economic activity as a simple value. Therefore, this study examines the historical changes in spatiotemporal patterns of crop production in the Texas High Plains from the perspective of geographical concentration and spatial autocorrelation. Specifically, this study attempts to answer the following research questions: (i) Does the pattern of geographical concentration of corn and wheat crop production show spatial trends and if so, do those trends change over time? (ii) Is there any spatial dependence in the production of major crops (corn and wheat) across the counties of the Texas High Plains?

From this research, regional producers and the public will benefit from the ability to access and visualize the spatial and temporal patterns of regional crop production activity information. Further, considering the spatiotemporal change of cropping patterns in the Texas High Plains, the results of this study may provide information to policy makers about where conversion to non-irrigated varieties may occur first as dependence on irrigation water from Ogallala Aquifer is reduced.

## 2. Materials and Methods

This section explains the study area, methods, and data sources for this study.

Methodology: This study examined the spatial and temporal changes in cropland acreage in the Texas High Plains from the perspective of geographical concentration and spatial dependence. First, to examine how much (or little) the acreage of cropland in the Texas High Plains has changed over time, maps were generated for each variable of interest. To create maps, categories were created which were dependent on the range of values. Therefore, the maps can be used to identify counties with an extremely large (or small) number of acres. Second, to analyze the overall change in geographical concentration, the Gini coefficient, one of the most commonly used measures of geographical concentration of industries, was calculated [9,18–21]. In this study, the geographical concentration of cropland acreage refers to the relative share of harvested acreage contributed by each county. In particular, the Gini coefficient was calculated as:

$$G = \left( \sum_{i=1}^{n} (2i - n - 1)x_i \right) / n^2 \mu \tag{1}$$

where $x$ is the number of acres harvested, $n$ is the total number of counties, $i$ is the rank of values in ascending order, and $\mu$ is the mean value of $x$. The Gini coefficient takes a value between zero and one. A value of zero means that each county harvests the same number of acres, while a value of one indicates that all of the production is concentrated in a single county.

Third, to examine the spatiotemporal changes in geographical concentration, size distributions were calculated for all variables of interest. To do so, the counties were first ranked in descending order based on the number of acres. The cumulative distribution of acres harvested was then generated. The number of counties with 25, 50, and 75 percent of total acres harvested was then determined. A map-based visual representation of geographical concentration for each census year was then created. Additionally, to investigate spatiotemporal changes in irrigation decisions, a quantile map was generated using data

on irrigated acreage as a percent of total acreage for each variable of interest and each census year.

Finally, to examine the spatial autocorrelation or dependence of crop choices across the study region, the Moran's I statistic was calculated [22]. Spatial autocorrelation is characterized by a correlation among nearby locations. Specifically, the Moran's I statistic measures how one county's spatial information content is similar to that of the surrounding counties. In this study, the statistical test was performed to test whether the relative proportion of each harvested crop was randomly distributed across counties. In this study, the relative proportion of each crop was computed as the ratio of the number of acres harvested for each crop to the total number of acres of cropland. The test was conducted using both total acreage and irrigated acreage figures. The Moran's I statistic ranges from −1 to 1. A value of −1 indicates perfect dispersion and perfect clustering of dissimilar values; a value of 0 indicates that there is no autocorrelation among the neighboring counties; and a value of 1 means perfect clustering of similar values. In other words, a higher value of Moran's I indicates that the observations are clustered near other high values relative to lower values [23]. The *p*-value of Moran's I index determines whether the null hypothesis of no spatial autocorrelation can be rejected.

*Data Sources*

Study Area: This study analyzes temporal changes in spatial patterns of crop production at the country level in the Texas High Plains. The study area includes the following 39 counties in the Northern and Southern High Plains Texas Agricultural Statistics Service (TASS) districts: Andrews, Armstrong, Bailey, Briscoe, Carson, Castro, Cochran, Crosby, Dallam, Dawson, Deaf Smith, Floyd, Gaines, Glasscock, Gray, Hale, Hansford, Hartley, Hemphill, Hockley, Howard, Hutchinson, Lamb, Lipscomb, Lubbock, Lynn, Martin, Midland, Moore, Ochiltree, Oldham, Parmer, Potter, Randall, Roberts, Sherman, Swisher, Terry, and Yoakum (Figure 1). The region is comprised mostly of agricultural land, with nearly 11.4 million acres of cropland in 2017 [24].

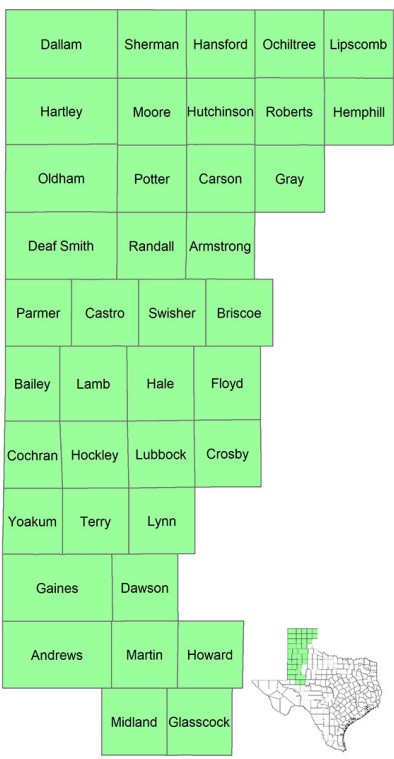

**Figure 1.** The 39-county area of the Northern and Southern High Plains Texas Agricultural Statistics Service (TASS) districts and their location in the state of Texas.

The climate is semi-arid with low rainfall. Annual precipitation varies widely across these 39 counties with long-term averages of 14.2–23.7 inches [25]. Due to low precipitation, crop production in the area is highly dependent on irrigation water from the Ogallala Aquifer. Given the depletion of the Ogallala Aquifer, it is important for policy purposes to examine historical changes in spatiotemporal patterns of crop production in the region. The Northern and Southern High Plains Texas Agricultural Statistics Service (TASS) districts and their location in the state of Texas is shown in Figure 1.

**Agricultural Census Data.** Historical county-level agricultural census data were collected for the years 1978, 1982, 1987, 1992, 1997, 2002, 2007, 2012, and 2017 from the United States Department of Agriculture (USDA) and National Agricultural Statistics Service (NASS). Two crops, corn and wheat, were selected for this study because these are the most prominent crops in the Texas High Plains and census data were available for these crops throughout the study period. The variables included in this study are total and irrigated harvested cropland area and number of farms for each selected crop. Census data were chosen for analysis because it presents a nearly complete, county-level enumeration of crop production data in the U.S. making it possible to examine spatial variations between countries over multiple time periods [10].

### 3. Results

#### 3.1. Corn Harvested for Grain

The total and irrigated harvested corn grain acres are presented in Figure 2. The total harvested corn acres decreased by approximately 9% from 799,000 acres in 1978 to 726,000 acres in 2017. From 1978 to 1987, both the total and irrigated harvested corn acreages declined fast and reached a minimum of 409,000 and 397,000, respectively. Overall, except for the years 2012 to 2017, both total and irrigated acres of corn harvested for grain changed in the same direction. This suggests a decline in the irrigated percentage of the region's corn acreage.

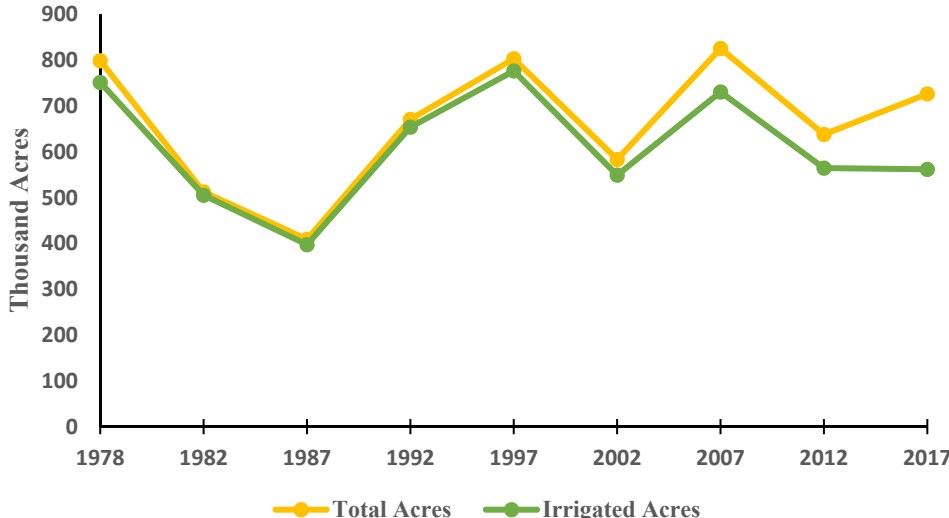

**Figure 2.** Total and irrigated corn harvested area for grain (in thousand acres) in the Texas High Plains, 1978–2017.

The maps for irrigated and non-irrigated acres of corn harvested for grain by county from 1978 to 2017 are presented in Figure 3. Parmer County had the largest corn acreage (more than 150,000 of harvested acres) in 1978 and 1982. Nevertheless, its total harvested corn acres gradually decreased over time. In 1987 and 1992, the counties with the largest share of corn acreage were Parmer, Hale, Castro, and Dallam. These counties had at least 50,000 and 75,000 acres of their cropland acres planted with corn in 1987 and 1992, respectively. From 1997 to 2012, Dallam had become the county with the largest corn acreage. In 2017, however, Sherman had become the county with the largest corn acreage.

A similar trend was found when focusing solely on the irrigated acres of corn harvested for grain. Specifically, Parmer County had the largest corn acreage from 1978 to 1992. From 1997 to 2012, Dallam County had the largest share of irrigated corn acreage, whereas Sherman took first place in 2017.

This study also estimated the Gini coefficients for total and irrigated acres of corn for grain for all census years. Table 1 explains the estimated coefficients of Gini for total and irrigated acres of corn for grain. The Table 1 results report that for total corn acreage, the Gini coefficient ranged between 0.636 and 0.790. For irrigated corn acreage, the Gini coefficient varied between 0.683 and 0.797. The high value of the Gini coefficients indicates that both total and irrigated corn acreages were consistently concentrated in a smaller number of counties over time.

**Table 1.** Gini coefficients of total and irrigated corn.

| Year | Total | Irrigated |
|---|---|---|
| 1978 | 0.756 | 0.773 |
| 1982 | 0.790 | 0.797 |
| 1987 | 0.749 | 0.753 |
| 1992 | 0.717 | 0.723 |
| 1997 | 0.715 | 0.724 |
| 2002 | 0.782 | 0.786 |
| 2007 | 0.742 | 0.746 |
| 2012 | 0.738 | 0.741 |
| 2017 | 0.636 | 0.683 |

Notes: Source: authors' own calculations. This table reports the estimated Gini coefficients for total and irrigated corn acres harvested for grain, 1978–2017.

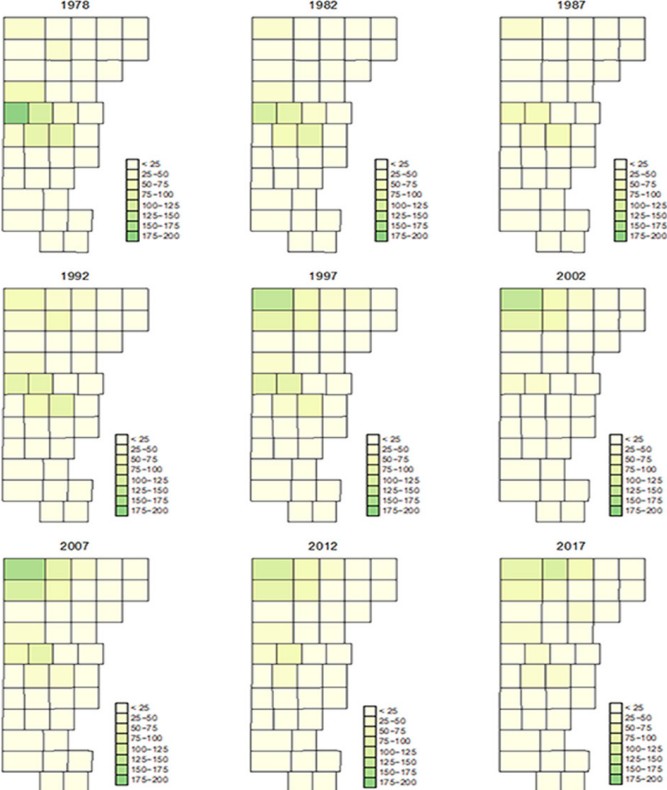

**Figure 3.** Irrigated and non-irrigated corn harvested area for grain (in thousand acres) by county, 1978–2017.

The geographical concentration of total corn acreage is presented in Figure 4. Overall, a total of two to three counties covered a quarter of total acres of corn harvested for grain: Parmer and Castro for the census years 1978 and 1982; Parmer and Hale for the census year 1987; Parmer, Hale, and Castro for the census year 1992; Dallam and Castro for the census year 1997; Dallam and Hartley for the census years 2002, 2007, and 2012; and Sherman and Dallam for the census year 2017. The combined land area in the top two or three counties covered less than 10 percent of the total land area. Overall, there was a small change in the geographical concentration of total corn acreage. A similar trend was observed when considering the irrigated corn acreage. These results are expected as most of the corn acreage was irrigated.

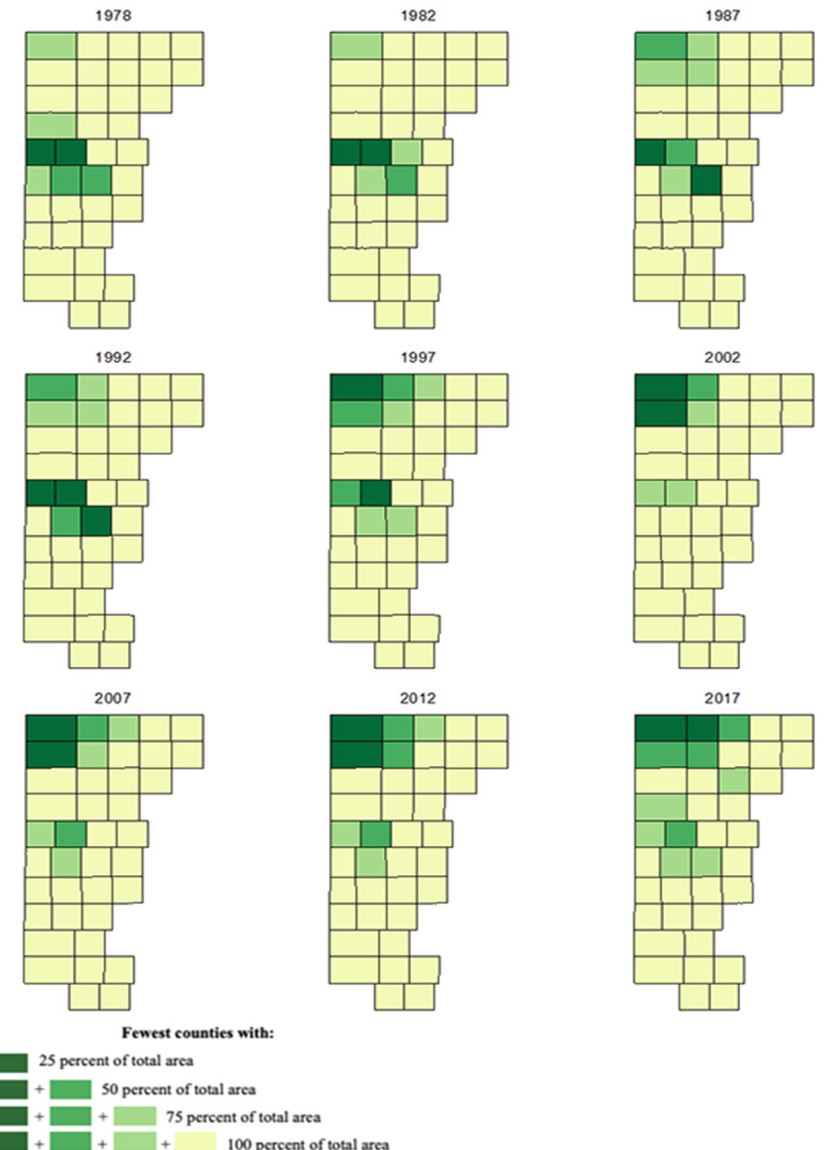

**Figure 4.** Concentration of irrigated and non-irrigated corn harvested area for grain (in thousand acres) by county, 1978–2017.

*3.2. Wheat Harvested for Grain*

The region's total and irrigated acres of wheat harvested for grain during the census years 1978 and 2017 are depicted in Figure 5. The total acres had been clearly much more volatile than the irrigated acres. The total acres decreased from 1.4 million acres in 1978 to 1.1 million acres in 2017, whereas the irrigated acres decreased from 595,775 acres in 1978 to only 250,723 in 2017.

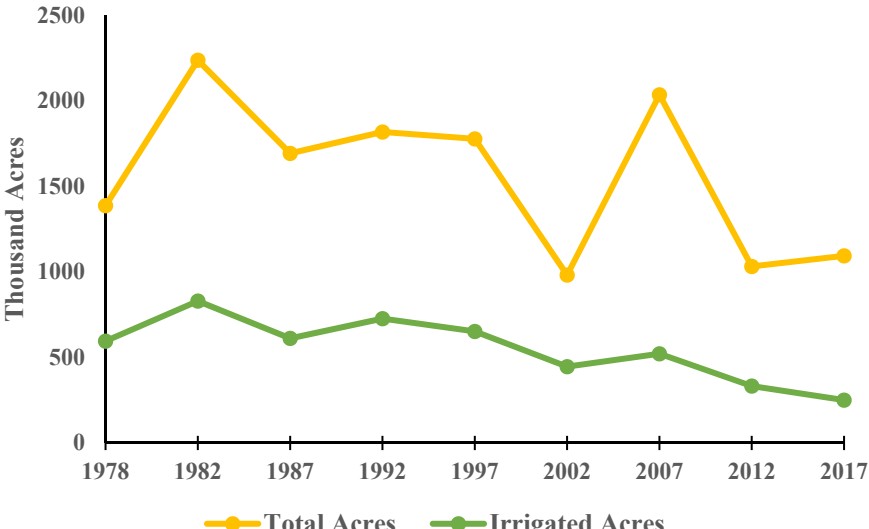

**Figure 5.** Total and irrigated wheat harvested area for grain (in thousand acres) in the Texas High Plains, 1978–2017.

Next, the maps of total and irrigated acres of wheat harvested for grain are illustrated in Figure 6. Overall, wheat production was concentrated in the northern part of the region. Focusing on total wheat acreage, Ochiltree had the largest number of acres in most census years. During the census year 1992, Ochiltree was considered wheat-dense (having more than 150,000 acres of wheat harvested). In the most recent census year, Deaf Smith was the county with the largest wheat acreage in the area. Similar patterns were observed when considering irrigated wheat acreage.

This study also estimated the Gini coefficients for total and irrigated acres of wheat harvested for grain for the census period 1978–2017. The estimated Gini coefficients for total and irrigated acres of wheat harvested for grain are reported in Table 2.

**Table 2.** Estimated Gini coefficients for total and irrigated wheat harvested for grain.

| Year | Total | Irrigated |
|------|-------|-----------|
| 1978 | 0.542 | 0.647 |
| 1982 | 0.485 | 0.586 |
| 1987 | 0.527 | 0.624 |
| 1992 | 0.522 | 0.632 |
| 1997 | 0.535 | 0.643 |
| 2002 | 0.488 | 0.619 |
| 2007 | 0.535 | 0.622 |
| 2012 | 0.539 | 0.570 |
| 2017 | 0.554 | 0.616 |

Notes: Source: authors' own calculations. Estimated Gini coefficients for total and irrigated wheat harvested for grain acres, 1978–2017.

The estimated Gini coefficients for total and irrigated acres of wheat harvested for grain are reported in Table 2. For total wheat acreage, the Gini coefficient value ranged between 0.488 and 0.554. For irrigated wheat acreage, the Gini coefficient varied between 0.570 and 0.647. Similar to corn crops, the results suggest that wheat acreage was concentrated in a small number of counties. As the Gini coefficient values for the case of irrigated acres were higher than those for the case of total acres, irrigated wheat acreage was more concentrated in fewer counties than total (irrigated and non-irrigated) wheat acreage.

Further, spatial changes in the geographical concentration of total and irrigated acres of wheat harvested for grain are shown in Figure 7.

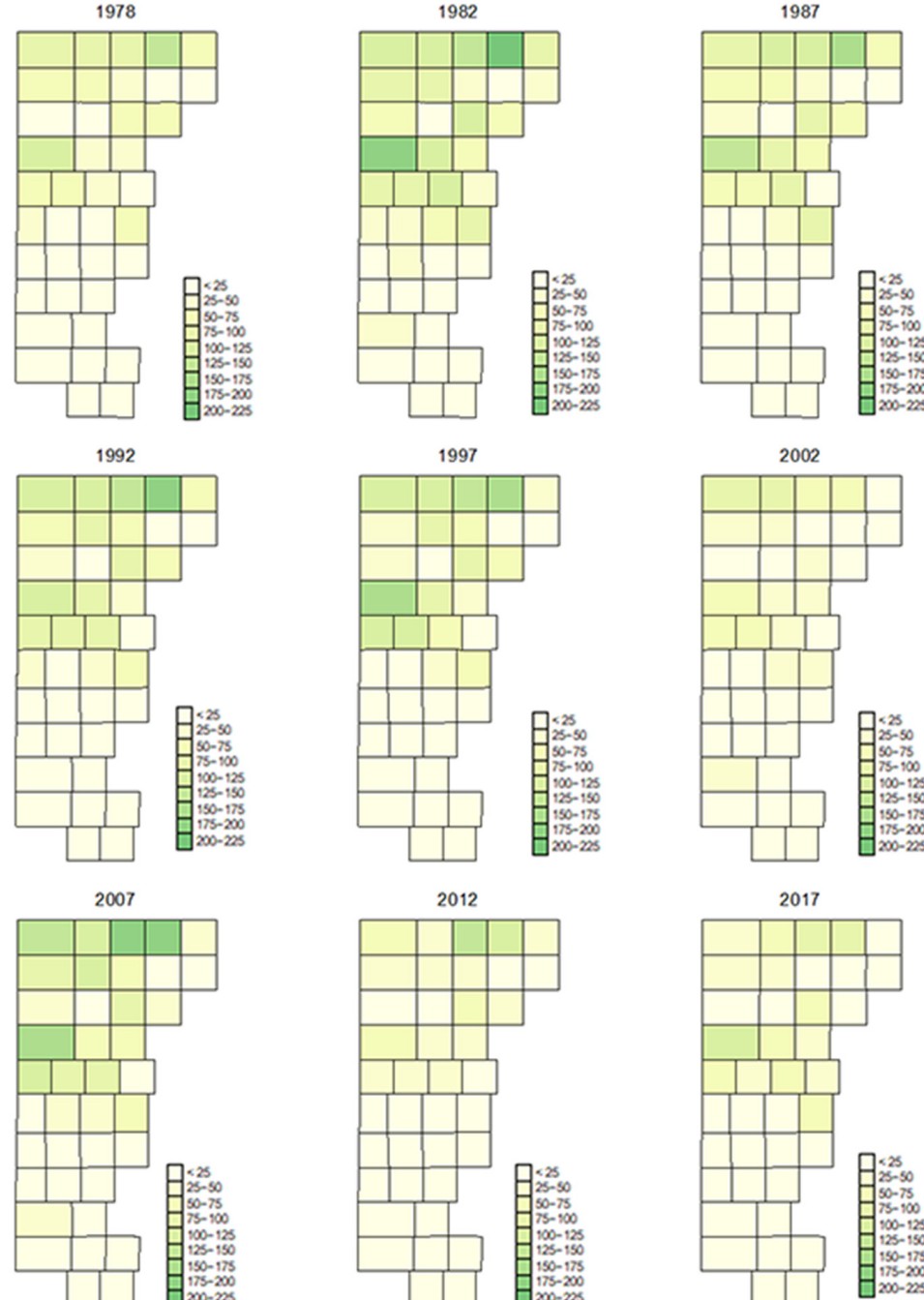

**Figure 6.** Irrigated and non-irrigated wheat harvested area for grain (in thousand acres) by county, 1978–2017.

As can be seen from Figure 7, the fewest number of counties with 25 percent of total wheat acres were two to four counties, depending on the census year. In 1978, the top three counties were Ochiltree, Deaf Smith, and Hansford. During the census years 1982 and 2017, six counties covered more than 25 percent of the harvested wheat acreage: Ochiltree, Deaf Smith, Hansford, Randall, Dallam, and Sherman. These counties, however, covered less than 15 percent of the total land area. Similar patterns were observed when focusing on the case of irrigated wheat acreage. However, these counties covered less than 15 percent of the total land area. Overall, wheat acreage was concentrated in the northern part of the region.

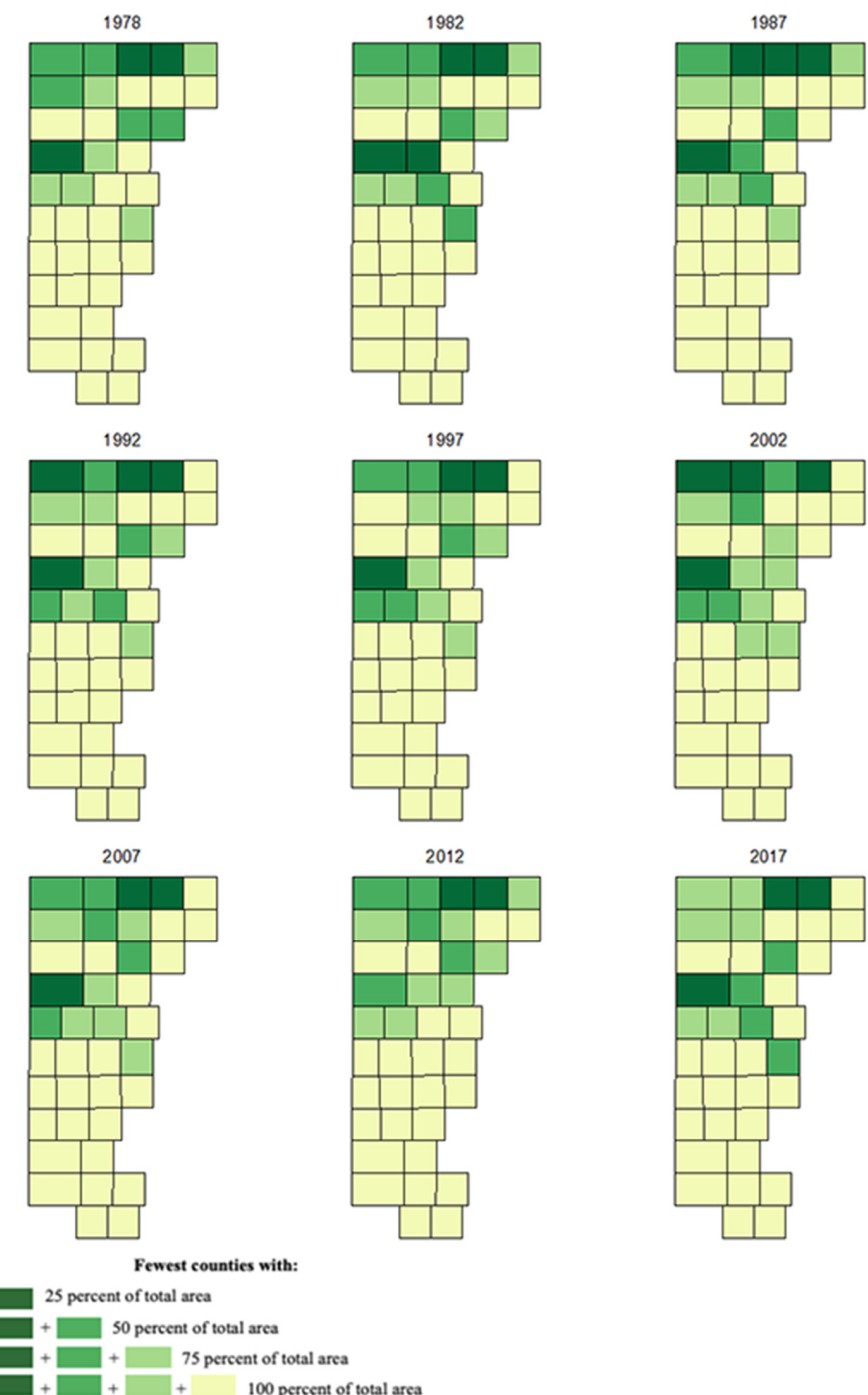

**Figure 7.** Concentration of irrigated and non-irrigated wheat harvested area for grain (in thousand acres) by county, 1978–2017.

### 3.3. Spatial Autocorrelation

To assess spatial autocorrelation or dependence in crop choices among neighboring counties, Moran's I statistics were calculated to test whether the relative proportion of each crop harvested was randomly distributed across counties. The test results for total acreage (irrigated and non-irrigated) are reported in Table 3. Overall, the Moran's I statistics were positive and statistically significant at the 5 percent significance level, indicating the existence of spatial dependence. The test statistic ranged from 0.421 to 0.670, when considering both irrigated and non-irrigated cropland area and from 0.448 to 0.690, when considering only irrigated cropland area. For wheat, it is clear that the Moran's I statistics

declined over time when considering the proportion of wheat relative to the total cropland area. In fact, the null hypothesis of no spatial autocorrelation cannot be rejected for the census year 2017. Nevertheless, when considering only irrigated cropland, the Moran's I statistics were consistently high for the case of wheat, indicating strong autocorrelation across neighboring counties. Overall, the spatial autocorrelation results are in line with the exploratory data analysis results presented in the previous sections.

**Table 3.** Estimated Moran's I statistics for total and irrigated acreage.

| | Total | | Irrigated | |
| --- | --- | --- | --- | --- |
| Year | Corn | Wheat | Corn | Wheat |
| 1978 | 0.551 * | 0.917 * | 0.503 * | 0.904 * |
| 1982 | 0.486 * | 0.753 * | 0.448 * | 0.907 * |
| 1987 | 0.518 * | 0.835 * | 0.491 * | 0.920 * |
| 1992 | 0.590 * | 0.922 * | 0.543 * | 0.919 * |
| 1997 | 0.625 * | 0.676 * | 0.599 * | 0.903 * |
| 2002 | 0.670 * | 0.518 * | 0.690 * | 0.824 * |
| 2007 | 0.638 * | 0.677 * | 0.581 * | 0.821 * |
| 2012 | 0.653 * | 0.170 * | 0.605 * | 0.754 * |
| 2017 | 0.421 * | −0.005 | 0.621 * | 0.744 * |

Notes: Source: authors' own calculations. * Denotes a rejection of the null hypothesis of no spatial autocorrelation at the 5 percent significance level.

## 4. Discussion

The total harvested cropland acres of wheat and corn were more volatile than the irrigated harvested cropland acres. The crop maps for both total and irrigated harvested acres of cropland for all variables of interest show that counties in the center of the Texas High Plains have the largest share of harvested acres and visualize the change in county-level harvested cropland over time. Further, 88 percent of the total land area of the study region is situated over the Ogallala Aquifer. Moreover, the counties in southern region of the Texas High Plains have a lower saturated thickness than the northern part. Therefore, these areas are not as affected by declines in the Ogallala Aquifer. Some other factors such as weather variables and soil quality might have some correlations which warrant further investigation.

As the Gini coefficient values for the case of irrigated acres were higher than those for the case of total acres, irrigated wheat acreage was more concentrated in fewer counties than total (irrigated and non-irrigated) wheat acreage.

Moreover, for quantiles of size distribution, total (irrigated and non-irrigated) and irrigated harvested acres of cropland corn and wheat were ranked from largest to smallest and the minimum number of counties with 25 percent, 50 percent, and 75 percent of total acres harvested were identified at the county level throughout the study period. There was almost no change in the number of counties with 25 percent of the total and irrigated harvested cropland acres concentration over the study period whereas there was a small change in the geographical concentration of total corn acreage where only two to three counties covered a quarter of the total acres of corn harvested for grain. However, wheat production was concentrated in the northern part of the region. The saturated thickness of the Ogallala Aquifer in the northern part of the Texas High Plains is higher (up to 500 feet than the southern or center part (up to 200 feet) which might drive a high concentration of wheat production in the northern region. Spatiotemporal changes in the proportion of irrigated harvested areas relative to total harvested area for each crop were also examined. Since corn is mostly an irrigated crop, more than 80 percent of the total corn acreage in most were irrigated in the Texas High Plains.

For corn, the Moran's I statistics are relatively stable over time when considering both irrigated and non-irrigated cropland area. However, for wheat, it is clear that the Moran's I statistics have declined over time when considering the proportion of wheat relative to the total cropland area.

## 5. Conclusions

This study examined the temporal changes in county-level spatial patterns of corn and wheat crop production in the Texas High Plains. Historical agricultural census data on acres of corn harvested for grain with acres of wheat harvested for historical census years were considered in the analysis. Total (irrigated and non-irrigated) crop acres were analyzed. The study also analyzed the spatial and temporal changes in corn and wheat acres in the Texas High Plains from the standpoint of geographical concentration and spatial dependence. Maps were generated for each variable of interest in order to examine how much cropland acreage in the Texas High Plains has changed over time. The temporal analysis results show that during the study period, the total and irrigated harvested acres showed a downward trend for almost all the variables of interest from 1978 to 2017.

To analyze the overall change in geographical concentration and spatiotemporal changes, the Gini coefficient and the quantiles of size distributions, respectively, were computed for all variables of interest. The high values of the Gini coefficients indicate that both total and irrigated corn acreages were consistently concentrated in a smaller number of counties over time. Similar to corn crops, the wheat acreage was concentrated in a small number of counties. Overall, considering the spatiotemporal change in corn and wheat cropping patterns in the Texas High Plains, policy makers should promote and support the non-irrigated varieties of crops to reduce the dependence on irrigation water from the Ogallala Aquifer.

**Author Contributions:** This research work titled "The Spatiotemporal Economic Analysis of Wheat and Corn Production in the Texas High Plains" was carried out in collaboration among all the authors. A.N. developed main idea of this study and was responsible for the majority of the writing. She also collected and organized the data to estimate the model and interpret the results. S.S. wrote introduction, specified the model, and wrote the conclusion and recommendations. L.K.A. supervised the overall write up throughout and B.G. reviewed, edited, corrected, and formatted this manuscript. All authors have read and agreed to the published version of the manuscript.

**Funding:** This research was funded in part by the USDA-ARS Ogallala Aquifer Program.

**Data Availability Statement:** Not Applicable.

**Acknowledgments:** We acknowledge the technical support of West Texas A&M University for providing access to materials pertaining to this research work.

**Conflicts of Interest:** The authors declare no conflict of interest.

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
