# Peer review of "Spatiotemporal Economic Analysis of Corn and Wheat Production in the Texas High Plains"

_water, doi:10.3390/w15203553_

Round 1
Reviewer 1 Report (Previous Reviewer 2)
The manuscript „The Spatiotemporal Economic Analysis of Corn and Wheat Production in the Texas High Plains” does not fall exactly in the scope of the journal, however there is some connection with water because irrigation of wheat and corn is very common in the study area. I have doubts if the manuscript should be submitted to the Water or other journal, eg. Agriculture should be better choice.
The manuscript demands some formatting adjustments because is not prepared exactly using the template, e.g. affiliations of the authors are not properly formatted, line spacing is not the same in all the manuscript, reference style is not proper, captions of the figures and tables have different formatting which is not proper.
The aim of the study presented in the end of the Introduction is not clear. What pattern do you mean? Do you mean grain yield, percentage of agricultural area, …? What do you mean as a “production” in the phrase “spatial dependence in the production”? Production is very general, usually it mean total quantity of grain per administration unit or other area. Please be more specific.
Fig. 3. The maps presents total area of corn. Since the area of the counties is not equal it would be better if percentage of the crop will be presented, not total area, because the same area does not mean the same percentage. In my opinion the analyses should be performed using two variables, i.e. percentage of corn in agricultural area and grain yield. Such two variables are two most important variables which characterize the crop production. I suggest to perform new analyses for these two variables, including calculation of spatial dependence, spatial autocorrelation.
Fig. 5 present values every 5 years. Could you present the results every one year?
Fig. 4 is without any legend and Fig. 7 has legend which is not compatible with the caption. It is necessary to improve these two figures. Please notice that tables and figures should be self-explanatory, ie. clear enough without reading all the text.
Section “Discussion” should be added to the manuscript and should compare the results with other similar studies.
Conclusions are very long. Conclusions should summarize the results not extend the results. I suggest to move some parts of the Conlusions to the Results and some parts to the Disciussion and leave only most important parts.
Author Response
Thank you for your valuable comments. It has improved our manuscript. Please see the attachment for our responses to each of your comments/suggestions. Thanks.
Lal Almas

Reviewer 2 Report (Previous Reviewer 1)
The paper can be accepted for publication in Water.
Author Response
Thank you for your support in publishing our research.
Round 2
Reviewer 1 Report (Previous Reviewer 2)
Authors improved the manuscript but not according all of my comments. Especially, one in my opinion very important comment (No. 4) was not addressed. Current version of the manuscript is better, however some changes are still necessary.
1. It is not necessary to include three times the same affiliation in the title page. Please follow the guidelines for authors.
2. In the abstract there are two types of fonts, ie. Palatino Linotype (correct) and TNR (not according the guidelines). Please be more careful in formatting the manuscript. The manuscript is not prepared exactly according the guidelines. I suggest to use template of the manuscript provided by the journal and follow all the guidelines.
3. Figures and tables are still not clear. For example Fig. 4 is without any legend. The figures and tables should be self-explanatory, i.e. clear enough without reading all the text.
4. References are not formatted according the guidelines.
The current version of the manuscript demands still deep changes. The changes made by authors are not sufficient and careless.
Author Response
Thank you for your valuable comments that has significantly helped us to improve our manuscript. Please see our response in the attached document.
Thanks.
Lal Almas

Round 3
Reviewer 1 Report (Previous Reviewer 2)
The manuscript was improved, but some important issues still not adressed. Section “Discussion” should be added to the manuscript and should compare the results with other similar studies.
Conclusions are very long. Conclusions should summarize the results not extend the results. I suggest to move some parts of the Conlusions to the Results and some parts to the Disciussion and leave only most important parts.
The manuscript is not exactly prepared according the guidelines, references are still not properly formatted, titles of the tables should be not in bold and other font type should be used. Please use the template of the manuscript for the journal.
Author Response
We have revised our manuscript according to the reviewers comments and suggestions. Please see our responses to their comments in the attached document. Thanks to all the reviewers for their valuable advice to improve our amnuscript.

This manuscript is a resubmission of an earlier submission. The following is a list of the peer review reports and author responses from that submission.
Round 1
Reviewer 1 Report
Submitted falls into the scope of water journal and I found it as interesting and relatively well written paper. The article is focused on The Spatiotemporal Economic Analysis of Wheat and Corn Production: Evidence from Texas High Plains
There are some issues with the manuscript which need to be addressed.
First of all, the manuscript should be carefully revised as far as the language is concerned. However, the manuscript's experimental design is good but unfortunately, not defined.
The manuscript pattern has a problem because the authors do not mention the line numbers.
In my opinion, the abstract needs to be written more scientifically, as in the results section, it mentions corn and wheat trends to decrease throughout the study period." Please add some (%) to how much the trend decreased. as well as what the pattern of results is based on the Moran’s-I test static. However, in the abstract "define Research Impact statement," I think it’s better to put this in the conclusion section.
The keywords are appropriate for the manuscript.
In the manuscript topic, first mention the wheat and corn production, and in the results section, first focus on corn and then define wheat. Please make one pattern throughout the manuscript; in the result section, first define the wheat production results and then corn. And if authors first want to define the corn results, please put in manuscript corn and wheat production.
In introduction section, last sentence “visualize changes in spatiotemporal patterns of crop production” please clearly mention which crops? and please clearly define the objectives of the study
In the methodology section, please add the technical route (road map of the study) where you clearly and step-by-step define the analysis
In my opinion, (figure 8, which is put in the conclusion part, is better put in the methodology section of the manuscript and then, based on its importance, also mentioned in the conclusion section.
Please add some more references that are related to and supported by the current study.
Minor English language need to be improved.
Reviewer 2 Report
The manuscript entitled „The Spatiotemporal Economic Analysis of Wheat and Corn Production: Evidence from Texas High Plains” presents interesting study on spatial evaluation of long-term profitability of crop production.
The manuscript is not prepared according the guidelines of the journal. For example Research Impact Statement is redundant ad should be removed, template of the document should be used, citation within the text should be presented using numbers not using names of the authors and years, sections of the manuscript should be according the guidelines of the journal.
The aim of the study presented in the end of the Introduction is not clear. What pattern do you mean? Do you mean grain yield, percentage of agricultural area, …? What do you mean as a “production” in the phrase “spatial dependence in the production”? Production is very general, usually it mean total quantity of grain per administration unit or other area. Please be more specific.
I suggest to merge Literature Review to Introduction or Material and Methods. This section should not be a separate section (see guidelines for authors).
Fig. 3. The maps presents total area of corn. Since the area of the counties is not equal it would be better if percentage of the crop will be presented, not total area, because the same area does not mean the same percentage. In my opinion the analyses should be performed using two variables, i.e. percentage of corn in agricultural area and grain yield. Such two variables are two most important variables which characterize the crop production. I suggest to perform new analyses for these two variables, including calculation of spatial dependence, spatial autocorrelation.
Fig. 5 present values every 5 years. Could you present the results every one year?
Fig. 4 and Fig. 7 present concentration of corn and wheat. In both maps many values are above 50% (even near to 100%). Sum of these two crops will be above 100%. It does not make sense. Could you explain why the values on the both maps are so high?
Section “Discussion” should be added to the manuscript and should compare the results with other similar studies.
Instead of the section Future Direction there should be section Conclusion which summarize the results obtained in the study.
References are not formatted properly.